# ZeOncoTest: Refining and Automating the Zebrafish Xenograft Model for Drug Discovery in Cancer

**DOI:** 10.3390/ph13010001

**Published:** 2019-12-24

**Authors:** Carles Cornet, Sylvia Dyballa, Javier Terriente, Valeria Di Giacomo

**Affiliations:** ZeClinics SL, IGTP Edifici Muntanya (c/de Can Ruti, Camí de les Escoles s/n; 08916 BDN, Lab P0-8), 08916 Barcelona, Spain; carles.cornet@zeclinics.com (C.C.); sylvia.dyballa@zeclinics.com (S.D.)

**Keywords:** zebrafish larvae, cancer, xenograft, drug discovery, automation, throughput

## Abstract

The xenograft of human cancer cells in model animals is a powerful tool for understanding tumor progression and metastatic potential. Mice represent a validated host, but their use is limited by the elevated experimental costs and low throughput. To overcome these restrictions, zebrafish larvae might represent a valuable alternative. Their small size and transparency allow the tracking of transplanted cells. Therefore, tumor growth and early steps of metastasis, which are difficult to evaluate in mice, can be addressed. In spite of its advantages, the use of this model has been hindered by lack of experimental homogeneity and validation. Considering these facts, the aim of our work was to standardize, automate, and validate a zebrafish larvae xenograft assay with increased translatability and higher drug screening throughput. The ZeOncoTest reliability is based on the optimization of different experimental parameters, such as cell labeling, injection site, automated individual sample image acquisition, and analysis. This workflow implementation finally allows a higher precision and experimental throughput increase, when compared to previous reports. The approach was validated with the breast cancer cell line MDA-MB-231, the colorectal cancer cells HCT116, and the prostate cancer cells PC3; and known drugs, respectively RKI-1447, Docetaxel, and Mitoxantrone. The results recapitulate growth and invasion for all tested tumor cells, along with expected efficacy of the compounds. Finally, the methodology has proven useful for understanding specific drugs mode of action. The insights gained bring a step further for zebrafish larvae xenografts to enter the regulated preclinical drug discovery path.

## 1. Introduction

Cancer is the second leading cause of death worldwide, accounting for 9.6 million estimated deceases in 2018 [1]. It is also the primary indication for pharmaceutical investment and medical care expenditure. During the last few decades, a better understanding of tumor biology has led to the development of therapies that increased the survival rate for multiple cancer types [2]. However, most common treatments consist of systemic administration of chemotherapeutics that target proliferating cells in neoplasms, but also in normal tissues with high regeneration capacity, causing severe side effects. Moreover, chemotherapies do not affect quiescent cells (i.e., cancer stem cells; CSCs), which may play an important role in drug resistance, relapse, and metastatic dissemination [3]. Finally, some types of cancer remain deadly because they are not responsive to available treatments. All of these drawbacks reflect the need for discovering more precise, safe, and efficacious therapies [1].

The drug discovery process in oncology is traditionally initiated by the use of fast in vitro systems for high throughput screenings. However, cell cultures are oversimplified tools due to the absence of tumor heterogeneity, microenvironment components, and anatomical structures for proper growth and metastatic potential evaluation [4]. In order to validate, and filter out in vitro results, drugs are tested in more biologically relevant in vivo models before entering clinical phases. Mouse xenografts of human cancer cells have become the prevailing translational tool in preclinical studies [5]. The use of rodents is recommended based on the high resemblance of transplanted tumors to the original ones. In addition, chick chorioallantonic membrane xenografts might be useful, although poorly validated, for studying metastatic events [6]. Despite their translational value, the high cost and low throughput of these models hamper their use for screening large numbers of possible therapies. Moreover, individual cancer genotype is emerging as a crucial factor leading therapeutic decisions. Thus, patient derived xenografts (PDXs) are potentially powerful tools, but the high amount of tissue needed for transplants in rodents is a major issue [7]. These flaws in the current preclinical models might partially explain the high costs and elevated drug attrition rate (95%) during the progression of anti-cancer treatments through the subsequent clinical phases [8,9]. Therefore, the use of innovative screening systems, including alternative animal models, could reduce costs and time and allow for assessing more compounds. This would indeed increase the chances of success in clinical trials and, hence, reduce drug attrition rates.

Zebrafish is an established animal model for academic research, but a more recent addition in the pharmaceutical drug development toolbox. However, its use is expanding as a fast and economical alternative to rodents for disease understanding, target validation, and drug discovery in multiple indications [10,11,12,13,14]. In regard to oncology, zebrafish displays a variety of features of a great translational value. First, it develops tumors if exposed to carcinogenic substances [15,16,17]. Second, oncogenes, tumor suppressors, and the main molecular pathways involved in cancer progression are highly conserved between zebrafish and humans [18]. In fact, embryos can be genetically manipulated for understanding the role of specific genes in diseases and, in particular, in tumor development [19,20,21]. Third, the immature larvae immune system allows transplantation and survival of human cancer cells with no need of immunosuppression [22,23]. In addition, it is important to note that all paracrine signaling pathways important in cancer development—WNT, EGF, FGF, VEGF, etc.—are well conserved and known to be involved in cancer progression in zebrafish [13]. The same occurs with the endocrine system, where androgen, estrogen, and thyroid systems are mature from 48 hpf, as implied by their response to canonical ligands and chemical disruptors [24]. Finally, larvae optical transparency and small size grant in vivo tracking of xenotransplanted fluorescent cells through standard and confocal imaging for measuring tumor growth and detecting early metastatic events, such as intravasation and extravasation, which are difficult to uncover in murine xenografts [25]. Based on these translational advantages, human tumor cells’ xenograft in zebrafish larvae have been implemented as a potentially useful tool for drug discovery [26,27,28]. The method conceivably allows for comparing cancer development with or without candidate therapies in just a few days, at a lower cost and higher throughput than equivalent murine experimental assays. Moreover, the low amount of material needed for the transplants makes the use of zebrafish larvae possibly amenable for PDXs and precision medicine [29,30]. Despite these benefits, previous reports have displayed contrasting results due to technical differences, mostly in terms of incubation times, image acquisition and analysis methods, cell labeling and site of injection. The lack of experimental standardization and validation might have led to a lower-than-expected exploitation of this screening tool in a pharma industry setting.

In order to increase drug screening throughput, while providing robust biological translatability, it has been established an optimized and standardized xenograft system in zebrafish larvae, the ZeOncoTest. To this end, different available procedures were reviewed, tested, and compared. As such, robust settings for cell labeling, injection into larvae, and automated imaging and image analysis were implemented. Finally, the accuracy of the method was validated by assessing the behavior of a battery of tumor cell lines and the impact of different compounds on their growth.

## 2. Results

Close to 250 articles focusing on the use of zebrafish larvae cancer xenografts were reviewed to define the best experimental approach (Appendix A). The majority of the studies were consistent on the cell injection timing, with 86% showing larvae injection at 48 h post fertilization (hpf). However, we detected a great variability in most of the other experimental settings and conditions. As such, the incubation temperature for xenotransplanted larvae ranged from 28.5 °C, standard for fish development, to 36 °C, closer to the ideal temperature for human cells growth of 37 °C. Individualization of animals throughout the experiment was only performed in 10% of the reports. The most common cell labeling methods were represented by fluorescent dyes, in 70% of the studies, compared to 30% using cells stably expressing fluorescent proteins. In addition, 82% of the dyes employed was membrane-specific. Among those, a chloromethylbenzamido derivative of octadecylindocarbocyanine (CMDiI) was utilized in 64% of the cases. As for the injection site, the yolk was the standard choice in half of the reports, followed by the perivitelline space (pvs) used in the 25% of the studies. Injections in the vasculature, brain ventricle, and heart were performed in the remaining 25% of the cases. Finally, the use of zebrafish larvae xenotransplants to address drugs mechanism of action (MoA) was only shown in half of the studies (Appendix A). Based on these data and our own expertise, we tested and compared different experimental conditions to define the most adequate settings for the ZeOncoTest.

### 2.1. ZeOncoTest: General Experimental Setup and Workflow

Basic conditions, such as stage of injection and incubation temperature, were set to guarantee the survival of animals and xenotransplanted human cancer cells. We chose to inject larvae at 48 hpf and to grow them at 35 °C. Those were the best settings reported in previous studies (Appendix A). In order to evaluate the anti-tumor effect of compounds, engrafted larvae were incubated across two time points between 0 and 144 h post injection (hpi), defined according to the growth and invasion capabilities of the different tested cell lines. The drug concentrations used were the No Observed Effect Concentrations (NOECs), determined through an acute toxicity test performed at the same conditions—stage, temperature—established for the subsequent efficacy assay. Finally, we determined to infer the anti-cancer effect of candidate drugs by comparing tumor growth and dispersion between untreated and treated xenografted animals (Figure 1).

### 2.2. Improved and New Methodologies

#### 2.2.1. 3D Imaging, Automation, and Throughput

A main source of potential bias in previous zebrafish xenograft reports resulted from estimating tumor areas from images acquired through stereomicroscopy or widefield microscopy. In our experience, when larvae were not equally oriented between different time-points, results based on area measurements were inconsistent (data not shown). To overcome this limitation, we decided to use confocal microscopy and acquire z stacks to extrapolate tridimensional images (volumes) of tumor masses at different time points. In our hands, the comparison of volumes across time provided a more reliable estimation of the growth of irregularly shaped tumor masses than the comparison of areas between non-equally oriented larvae, as exemplified in Video S1.

In addition, 90% of the previous reports measured tumor progression through comparing the average of whole populations between time points (Appendix A). An important improvement is represented by the fact that we chose to keep the animals individualized in 96 well plates during the procedure. This approach allows for comparing images from the same individual at different times and to measure single tumor growth and invasion. In contrast to group quantifications, singular estimations have an enormous impact on the statistical robustness of the method, since every larva can be considered a biological replicate. Moreover, we elected to analyze volumes ratios between time points, instead of absolute values, allowing the normalization with respect to the original tumor size. The outcome of the combination of these approaches results in a lower variability and, hence, a more accurate quantification.

To further increase the screening throughput, we developed a novel imaging system in 96-well plates through an automated microscopy platform. Finally, we developed an in-house automated image analysis pipeline to streamline the quantification of tumor volume and cell dispersion parameters in the same animals across different time points. The implementation of this workflow allowed the screening of 80 individualized fish per experiment, a throughput that, to our knowledge, has never been reached before.

#### 2.2.2. Choice of a Suitable Cell Labeling Method

In order to track cancer cells in grafted larvae, it was necessary to consider different factors regarding their labeling. First, we needed to select a method not causing toxicity throughout the duration of the experiment. Moreover, labeling was required to be homogeneous. Finally, we had to ensure a rapid loss of fluorescence following tumor cell death. The achievement of these purposes would guarantee a reliable evaluation of tumor volumes and dispersion across time and under different drug treatments.

As mentioned above, the use of fluorescent dyes is a common method to label cells prior to injection, as shown in 70% of the previous zebrafish larvae xenografts studies (Appendix A). Based on this, we tested the most widely used lipophilic membrane dye CMDiI and cytoplasmic dye Carboxyfluorescein succinimidyl ester (CFSE) in the breast cancer line MDA-MB-231. Both dyes were used at three concentrations and toxicity was detected at 2 μM for CMDiI and at 5 μM for CFSE (Figure 2A,B). At non-toxic concentrations for CMDiI (1 μM) and CFSE (2 μM), we detected non-homogeneous labeling (Figure 2C,D), although CMDiI displayed better retention than CFSE.

Once we defined the maximum tolerated concentration of the dyes, we proceeded to test fluorescence loss following cell death. As a positive control, we used MDA-MB-231 stably infected with a vector coding for the green fluorescent protein (GFP). It is known that physiological protein degradation, and resulting loss of GFP fluorescence, occurs following cell death. These fluorescent cells were labeled with 1 μM CMDiI and exposed to a high dose of Dimethyl sulfoxide (DMSO) to achieve cell death. In order to detect dead cells, 4′,6-diamidino-2-phenylindole (DAPI) staining was performed after 24 h. Surprisingly, 95.7% of DAPI positive dead cells still displayed CMDiI labeling retention, whereas only 5.4% of them kept GFP expression (Figure 2E). This striking outcome was confirmed after the measurement of tumor masses injected in zebrafish larvae. For this purpose, the yolk was chosen as an injection site for simplicity. As previously shown in other xenografts settings, we expected cell death to be evident shortly after transplantation, with only the most resistant cells adapting to the new microenvironment and then expanding [31,32]. The average tumor mass variation between the two time points was 9.9, taking into account CMDiI labelled cells, and 0.6 by analyzing GFP expressing ones (Figure 2F,G). CMDiI positive tumor masses were significantly larger than GFP positive ones in the same xenografts at the second time point, suggesting dead cells retained the dye staining, while they lost GFP protein expression, in accordance with the previous in vitro observations. Both sets of data in vitro and in vivo indicate that the use of membrane (CMDiI) and cytoplasmic (CFSE, data not shown) dyes does not allow a proper discrimination and exclusion of dead cells from the tumor mass imaging and analysis. Therefore, their use results in a biased over-estimation of tumor masses and growth.

Based on these new observations, the dyes commonly used in 70% of previous zebrafish larvae xenografts reports were discarded from our procedure. Then, stable infection with retroviral vectors coding for fluorescent proteins was chosen as the most suitable method to label tumor cells before transplantation. Indeed, fluorescent proteins’ expression guarantees non-toxic, steady, and homogeneous labeling, fluorescence transmission to daughter cells, and its dynamic loss at death by physiological protein degradation. As such, this option most probably warrants a proper tracking of grafted cells and an accurate estimation of tumor volumes and dispersion across different time points [33,34,35].

#### 2.2.3. Establishment of an Appropriate Injection Site

As mentioned before, half of the previous studies reported tumor cell injections in the yolk and 25% in the pvs (Appendix A). The main advantage of injecting intra-yolk is the accessibility. On the other hand, the yolk is a syncytium that might not provide an ideal microenvironment for the attachment and growth of solid tumor cells. According to our experience, transplantation in the pvs requires greater technical skills than injection into the yolk. This might be a reason behind the lower number of xenografts reported in this site, despite the fact that the pvs displays better features for anchorage-dependent cell growth and greater accessibility to the vascular system. Both aspects are likely to be crucial in primary solid tumor growth and metastases occurrence.

Following this rationale, we proceeded to test different cell lines to validate if solid tumors progression was favored in the pvs, when compared to the yolk. As expected, the highly proliferative tumor cell line MDA-MB-231 displayed a lower growth rate in the yolk, in comparison to the pvs, measured as an GFP-positive tumor mass fold change between tp1 and tp2 (1.5 vs 3.7 average, respectively; Figure 3A,B). Another aspect to consider was that growth impairment in the yolk was due to a progressively reduced space resulting from yolk consumption during larval development, which leads to an almost complete absorption by 96 hpi (144 hpf) [36]. This characteristic also implies that the experimental window for yolk xenografts is limited and shorter than the one for injection studies in the pvs. As such, colorectal cancer cells HCT116, whose proliferation rate was slowed due to the lower-than-physiological incubation temperature applied (35 °C), did not grow at the early tp2 (etp2) of 96 hpi, neither in yolk nor in pvs (0.9 and 0.5 tumor mass fold change average respectively). However, when injected in pvs, HCT116 cells could be followed until a late tp2 (ltp2) of 144 hpi, when tumor masses displayed a significant 1.7 fold increase in average (Figure 3C,D). Moreover, we did not observe metastatic events in larvae injected in the yolk with the highly invasive MDA-MB-231 and HCT116 tumor cell lines. On the contrary, metastases were detected in 76,2% and 54,5% of larvae transplanted with MDA-MB-231 and HCT116 cells in the pvs, respectively (Figure 3E–H).

To confirm the suitability of the pvs for xenotransplantation studies, we injected BJ non-transformed human fibroblasts, as negative experimental control. As expected, these cells did not grow or disseminate (Appendix A). This observation gives strength to the assumption that growth and invasion through the circulation were consequences of the tumorigenic behavior of transformed cells, and not artifacts due to the injection site.

Based on the direct comparison between injection sites, we have established the pvs as the optimal site of injection of cells to study drugs’ impact on primary tumor growth and dissemination.

### 2.3. Pharmacological Validation of the ZeOncoTest

We described above the methods that, through experimental support, provided the most robust strategy for addressing tumor growth and metastatic potential in the zebrafish larvae xenograft model. Once these improved methodologies were integrated into a single experimental workflow, the ZeOncoTest, we proceeded to its validation in drug discovery. To this end, we treated with different compounds zebrafish larvae transplanted with cell lines known to respond specifically to them. ROCK kinase inhibitor RKI-1447 was used to treat MDA-MB-231 larvae xenografts, since this drug has been shown to reduce growth and invasion of these cells in vitro and in mice models [37,38]. HCT116 transplanted animals were treated with docetaxel, as it was demonstrated that this molecule induces cell death in this cell line in culture and in mouse xenotransplants [39,40,41]. Finally, mitoxantrone was employed for the treatment of PC3 prostate cancer cells engrafted larvae, given that its use has been reported to cause cytotoxicity in these cells in previous in vitro and mice studies [42,43].

As hypothesized, the reported effects of the drugs on their selected target cells were reproduced in the ZeOncoTest. When compared to the DMSO-treated negative control population, RKI-1447 significantly reduced MDA-MB-231 tumor growth from 3.7 to 2 fold increase (Figure 4A and Appendix A). Docetaxel caused a decrease in HCT116 tumor mass (fold change average of 0.3), as opposed to an increase (fold change average of 1.9) observed when larvae were exposed to DMSO control (Figure 4B and Appendix A). Finally, PC3 volume expansion decreased from 3.3 to 1.6 fold, when comparing injected larvae treated with mitoxantrone to the DMSO-treated population (Figure 4C and Appendix A). Furthermore, we confirmed published data on RKI-147 activity towards abolishing the metastatic invasion capability of MDA-MB-231 cells. The comparison between the variances of the secondary tumor masses at tp2 between RKI-1447- and DMSO-treated larvae was 3.4 × 10^4^ versus 2.5 × 10^5^ (Figure 4D and Appendix A). These data offer a definitive proof of principle, more extensive than any other study before, for the suitability of the ZeOncoTest in drug discovery applications.

### 2.4. Addressing Drugs Mechanism of Action with the ZeOncoTest

We have shown that the ZeOncoTest can be a novel tool for addressing tumor growth and metastatic potential in a relevant drug discovery setting. To further explore its applicability, we decided to study if it could be used for answering more precise questions, such as the evaluation of anti-cancer drugs MoA.

In order to test this, we chose docetaxel MoA. The activity of this compound consists of disrupting the normal function of microtubules, thereby it stops mitotic division and induces death specifically in proliferating cells [44,45]. We performed equivalent experiments in vitro in cell culture and in vivo in xenografted zebrafish larvae. At first, HCT116 cells were incubated in culture with mitomycin C, which impairs cell proliferation and, subsequently, with docetaxel. As expected, docetaxel had no effect on the viability of the cells when proliferation was blocked by mitomycin C. On the contrary, it did have a significantly strong cytotoxic effect on proliferating cells not previously incubated with mitomycin C, but with phosphate-buffered saline (PBS) control, and a 92% decrease in cell number was detected (Figure 5A). Next, we tested the same experimental paradigm in our xenograft model. Interestingly, for HCT116 transplanted cells, a significant reduction in tumor growth (87%), in response to docetaxel treatment, was only observed at the late time point 2 (ltp2), when the cells do proliferate, as reported before. The cytotoxicity effect of the compound was not detected at an earlier time point 2 (etp2), when cells do not show detectable growth (Figure 5B). This result proves that our experimental model can also be used for addressing specific new anti-cancer candidate drugs MoAs.

## 3. Discussion

In the context of a growing cancer incidence, the search for biologically relevant, faster, and more affordable methods to discover new antitumoral drugs represents a medical priority. In order to increase the chance of finding novel effective drugs, zebrafish is imposing as an in vivo rational bridge between in vitro cell culture systems—cost-efficient and high throughput, but poorly predictive—and in vivo mammal models—more predictive, but expensive and time-consuming. Indeed, the final aim of the ZeOncoTest is to help choosing the best candidates, previously selected from in vitro experiments, to be subsequently tested in mice. This proposed pipeline would hopefully lead to a better clinical outcome and a lower attrition rate.

To guarantee zebrafish larvae healthiness, together with the growth of transplanted tumor cells, we selected the best general xenograft conditions. These were based on previous reports and our own intensive experimental optimization. As for the majority of previous studies, we chose to perform cell injections at 48 hpf, a stage at which the larvae are anatomically developed, but the immune system is not mature [46]. The first feature allows the study of both primary tumor growth and metastatic dissemination; the second aspect guarantees no rejection of transplanted cells from the host. Another reason for electing 48 hpf was the ample experimental time window: transplanted cells have enough time to grow before the larvae enter the juvenile stage and can no longer be kept in a laboratory environment. In addition, 35 °C was selected as the best compromise between the standard laboratory temperature used for zebrafish larvae to develop (28.5 °C) and the one required for human cells to grow appropriately (37 °C). This condition was also chosen in several previous studies [47,48,49]. Besides these basic features, additional technical aspects were reviewed and optimized to generate a standardized and robust assay. Stable expression of fluorescent proteins was chosen over staining with fluorescent dyes, as a more plausible solution for reflecting cell dynamics in terms of growth and death, based on our own results. As stated before, in direct cell labeling methods, the probe is present and detectable even after cell death, while the expression of a reporter gene is assumed to be lost, as described before [33,34,50]. However, although transgenesis is a suitable method for immortalized tumor cells, it might be technically difficult to apply to patient samples for the development of PDXs. To overcome this constraint and broaden the applicability of zebrafish larvae xenografts, there is a pressing need for the search of efficient patient samples labeling methods, which faithfully recapitulate tumor cell growth and survival dynamics. As for the site of injection, the pvs was directly compared and finally chosen over the yolk. The yolk is a viscous syncytium that provides a cell suspension-like environment not ideal for anchorage-dependent growth, whereas the pvs furnishes tissue support for solid cancer cells attachment and proliferation. In addition, the yolk gets consumed by 96 hpi (144 hpf), a characteristic that limits tumor growth until that time point. Finally, in the pvs, cells have an easier access to the vasculature than when injected into the yolk. This last feature allows the study of metastatic capability. In conclusion, our results demonstrate that transplantation in the yolk and the use of cell dyes provide a poor experimental setup to evaluate tumor progression. As for the choice of time points for the analysis, it was decided to generally set tp1 at 24 hpi, to allow injected cells to adapt to their new in vivo environment. However, highly invasive cells, such as MDA-MB-231, already showed metastatic dissemination at that stage (data not shown). Thus, tp1 was set at 2 hpi for cells displaying such behavior. In addition, 96 hpi was chosen as tp2 for highly proliferative cells, such as MDA-MB-231 and PC3. For cells showing a reduced growth rate at 35 °C, such as HCT116, tp2 was set instead at 144 hpi. This is a decisive aspect, given that these cells only underwent significant expansion if allowed to this latest time point. Importantly, cell expansion could be reached only in the case that injections were performed in the pvs, as previously explained. Then, the implementation of novel automated confocal imaging and analysis tools of individual larvae provides a potential screening throughput of dozens of conditions per month. Moreover, the chosen imaging method allows a much more exact estimation and comparison of tumor masses at different time points, through calculation of volumes instead of areas. It is important to note that measurement and correlation of areas are only recommended if the same orientation is achieved for each larva at the different time points. From our experience, distinct orientation among time points might lead to inaccuracy in areas calculation due to the irregular shape of tumor masses. We are now implementing a method based on the high-throughput microfluidic imaging system VAST [51] to allow equal positioning at each time point. This approach would allow a more accurate estimation of tumor size through area measurements. The main advantage of this further improvement would be the increase in screening throughput and the further simplification of image acquisition and analysis. Finally, in order to validate the ZeOncoTest, we tested the effect of known drugs on different cancer cell lines. The results provide a solid proof of principle of the strength of our methodology for addressing tumor growth and metastatic potential of different cancer cells. The choice of these three common and clinically relevant tumor types (breast, colorectal, and prostate cancers) proves the versatility and validity of our system for drug screening in solid tumors. We finally outlined the utility of our assay by evaluating anti-cancer drug efficacy and understanding their mode of action.

As for the study of metastatic progression, zebrafish, as well as rodents, provide a complex biological context that allows the assessment of tumor interaction with the microenvironment and the vasculature for the colonization of distant tissues. Nonetheless, metastasis evaluation in mice is mostly performed by bioluminescence imaging and tissue dissection followed by histopathology. These approaches take months and mostly address the latest colonization events [52,53,54]. Zebrafish larvae transparency allows tracking of transplanted fluorescent cancer cells through live imaging. This feature offers the possibility of investigating early metastatic steps, such as intravasation and extravasation. Hence, the use of zebrafish larvae xenografts can be complementary to the use of rodents for understanding different aspects of metastatic progression.

Given the effort placed in the setup of the ZeOncoTest, yet there is a number of aspects that will require further research to be fully addressed. An important question is how suitable is this method for evaluating biologics instead of small molecules. A big advantage deriving from the use of zebrafish larvae in drug discovery is that small molecules can be administered in the incubating water, from where the larva absorbs them. In our experience, passive diffusion does not work for biologics. Thus, the administration route has to be implemented for such macromolecules, either by injection into the vasculature or by co-injection with tumor cells. Another aspect to consider is that the pvs provides a good microenvironment for cancer cells’ attachment. As previously mentioned, this context favors the growth of solid tumors. On the other side, given their preferential growth in suspension, pvs might not benefit the growth of leukemic cells [55]. Hence, an option could be the test of a different injection site, i.e., the yolk or the vasculature for liquid tumors to develop. Finally, an important trend in cancer drug discovery is the search for immunotherapies. This is an aspect difficult to address with our model. Although the zebrafish immune system is well conserved in terms of genetic markers, cell types and functions, our assay is performed at the larval stage, when the adaptive immune system has not been fully developed [56]. In order to investigate possible immunotherapies, it would be necessary to adapt the ZeOncotest methodology to later developmental stages, after 6–8 weeks, when the immune system is mature. This said, standard chemotherapy is still the main option for patients that are not sensitive to immunotherapy [57]. This is especially important considering that responses to immunotherapy in the US are as low as 12.46% [58]. Moreover, it has been shown that immunotherapy is more effective in association with chemotherapy [59,60,61]. Hence, the discovery of safer and more efficacious chemotherapeutic drugs, to be applied on their own or in combination with immune therapies, is still mandatory.

To conclude, it is important to mention that, besides addressing tumor growth and metastasis, the ZeOncoTest could be applied for understanding the impact of drugs on other cancer hallmarks. Thus, angiogenesis could be evaluated by injecting tumor cells into transgenic zebrafish lines with fluorescent vasculature, as shown in previous works [62,63,64,65,66]. The assessment of vessels sprouting and morphology changes after injections and drug treatment would give an estimation of new vessels formation in the vicinity of the tumor. Fluorescent transgenic lines could also be used for studying the role of inflammation and the dynamics of innate immune cells in response to the injected cancer cells [67,68,69]. Moreover, cells can be genetically modified prior to injection and/or could be injected into newly generated zebrafish mutant/transgenic lines, allowing for studying the function of genes either in the tumor or in its microenvironment. Finally, tumor cell subpopulations, such as cancer stem cells, could be specifically labeled to allow their tracking across time. This way, the precise effect of novel compounds on those cells can be studied. All of these features open a broad range of possible applications and further outline the advantages of applying the ZeOncoTest in cancer drug discovery and target validation.

## 4. Materials and Methods

### 4.1. Cell Lines and Zebrafish Handling

All cell lines were obtained from collaborators: MDA-MB-231 breast cancer cell line from Simon Schwartz Navarro (VHIR), HCT116 colorectal cancer cell line from Melinda Halasz (UCD), PhoenixA retroviral packaging cells, and PC3 prostate cancer cells from Bill Keyes (IGBMC) and BJ non-transformed foreskin fibroblasts from Maria Aurelia Ricci (CRG). They were cultured in Dulbecco’s Modified Medium (DMEM, BE12-614F, Lonza, Cultek, Basel, Switzerland) implemented with 10% Fetal Bovine Serum (FBS, 10270106, GIBCO, Termofisher, Waltham, MA, USA) 1% L-Glutamine (BE17-605E, Lonza, Cultek, Basel, Switzerland) and 1% Penicillin/Streptomycin (DE17-602E, Lonza, Cultek, Basel, Switzerland), and kept at 37 °C with 5% CO_2_ in a humidified incubator.

Adult Casper fish, obtained from the European Zebrafish Resource Center, were grown at 28.5 ± 1 °C in a 14:10 h light:dark cycle in a recirculating tank system. Embryos were obtained by mating adult fish through standard methods [70] and kept in an incubator at 28.5 °C until 2 days in and at 35 °C just after injections until the end of the experiment.

This study was performed under the ethical approval code 10567, provided by the Generalitat of Catalunya.

### 4.2. Infection and Dye Staining of Human Cell Lines

Phoenix A were transfected with a MSCV-GFP-Puro vector through the calcium phosphate method. After 48 and 72 h, the supernatant containing the retroviral particles was collected and filtered through a 22 μm filter (SLGP033RS, Merck, Readington, N.J., U.S.A.). HCT116, PC3, MDA-MB-231, and BJ cells were subjected to 2 rounds and 1 round of incubation with the supernatant containing the retroviral particles plus 8 μg/mL of polybrene (H9268-5G, Sigma-Aldrich, Saint Louis, MO, USA), respectively at the 2 time points. In addition, 48 h after the last round of infection, cells were selected using 2 µg/mL puromycin (P8833-10MG, Sigma-Aldrich, Saint Louios, MO, USA) and left to recover for 2 days. Finally, cells were detached, washed, and resuspended in Phosphate Buffered Saline (PBS) with 5% FBS. The brightest GFP+ cells were sorted by flow cytometry and kept in culture.

For dye staining, cells were detached with trypsin-versene EDTA 0.25% (H3BE17-161E, Lonza, Cultek), washed in PBS and incubated with CMDiI and CFSE at the concentrations of 1, 2, and 5 µM in PBS, respectively, for 5 and 20 min at 37 °C, followed by 15 min at 4 °C for CMDiI. Cells were finally washed twice with PBS 10% FBS.

### 4.3. Induction of Cell Death and Hexosaminidase Assay

For hexosaminidase assay [71], 6000 cells were plated in 96 well plates: 2 replicates of 3 wells each were done for every condition studied. Moreover, 24 h later, media was removed from 1 of the replicates per condition and cells were washed with PBS twice. In addition, 60 μl of substrate solution was added for an hour at 37 °C. Afterwards, 90 μl developer solution was added before recording absorbance at 410 nm. These same passages were then repeated at 96 h. The number of cells was extrapolated through a calibration curve, obtained through plating the cells in numbers of 6000, 12,000, 24,000, 48,000, and 96,000 and performing the hexosaminidase assay 24 h later, just after attachment. Cell death was induced by o/n exposure to 10% DMSO.

### 4.4. Zebrafish Injection

In addition, 48 hpf larvae were manually dechorionated and anesthetized by immersion in 0.48 mM tricaine methanesulfonate (A4050, Sigma-Aldrich, Saint Louis, MO, USA) in E3 medium. Around 200–400 cells were injected in the perivitelline space using standard micro-injecting instrumentation. An hour later, larvae were screened at the stereomicroscope to discard non-injected larvae and those injected in a non-specific site or with tumor cells already in circulation.

### 4.5. Drug Treatment

Prior to the incubation of injected larvae with the different selected drugs, the Non-Observed Effect Concentration (NOEC) was established (Table 1 and Appendix A). Sixteen larvae per treatment were exposed individually to at least 5 concentrations of the compound of interest, at the same conditions as in the ZeOncoTest: from 48 to 192 hpf at 35 °C in 96 well plates. Each larva was analyzed for mortality, body deformity, scoliosis, yolk size, heart edema, heartbeat, and movement at 24 and 144 h post incubation (hpi). The NOEC was calculated as the highest concentration at which both mortality and teratogenic scores were below 20% at 144 hpi, and then used for the treatment of the injected larvae. DMSO was used as the negative control. In parallel, cells in culture were treated with the NOEC of the compound to ensure its chemotherapeutic effect in vitro (data not shown).

### 4.6. Automated Confocal Imaging and Analysis

In order to enable lateral positioning of injected larvae and restrict the area to be imaged, 3D printed orientation molds [72,73] and low melting agarose (8092.11, Conda, Pronadisa, Madrid, Spain) were used to shape the wells of 96 well plates (265301, Thermo Scientific, Nunc, Waltham, MA, USA). Injected larvae were anesthetized by immersion in 0.48 mM tricaine methanesulfonate in E3 medium and transferred in the wells. Xenografts were imaged at 2 different timepoints using a Leica True Confocal Scanning-Spectral Photometric 5 (TCS SP5) inverted confocal microscope system with an automated plate and the Matrix Screener software. Due to different growth and invasion capabilities of distinct cancers, we defined specific experimental time frames for every tumor cell type of interest, by injection, and live imaging observation until a maximum of 144 h post injection (hpi). Growth and invasion were measured by comparing cell masses volume and dispersion between tp1 and tp2 for each individual larva.

Images were processed and volumes and dispersion of cells evaluated using in-house macros and the FIJI Is Just ImageJ (FIJI) software. Finally, tumor growth was calculated as the ratio between the volumes at tp2 and tp1 for each individualized larva. Cell dispersion was assessed with the Statistical Package for Social Science (SPSS) software (IBM). A minimum of 16 animals were used per condition, and 1 or 2 experiments were performed.

### 4.7. Statistical Analysis

Results were analyzed using GraphPad Prism v 7.04 (GraphPad Software Inc., La Jolla, CA, USA). Prior to the analysis, the ROUT method (Q = 1%) was applied to identify outliers and remove them. Then, the D’Agostino-Pearson omnibus normality test was used to assess if data were normally distributed. For 2 groups’ comparison, Student’s *t*-test was used when values were normally distributed, whereas Wilcoxon or and Mann–Whitney tests were used respectively for paired and unpaired non-parametric data. Statistical analysis for multiple comparisons was performed using One-Way ANOVA, followed by Tukey or Sidak tests, for data with a parametric distribution. A Kruskal–Wallis test was instead performed for multiple comparisons of non-parametric values. Differences were considered statistically significant when *p* < 0.05. In figures, 1 asterisk (*) indicates *p* < 0.05, 2 (**) means *p* < 0.01, 3 (***) signify *p* < 0.001, and 4 (****) stays for *p* < 0.0001. No statistical significance is indicated by “ns”. Results are presented as mean ± standard deviation (SD).

## 5. Conclusions

The ZeOncoTest standardizes, automates and validates the zebrafish larvae xenograft system as a bridge between cell culture methods and mice models in cancer drug discovery. We propose it as an intermediate reliable assay for choosing the best compounds, previously selected from *in vitro* experiments, to be subsequently tested in rodents. Such pipeline would hopefully lead to a better clinical outcome and a lower attrition rate. Moreover, we show that the use of zebrafish larvae xenografts can be complementary to mouse studies for understanding early phases of metastatic progression. Our results move the zebrafish larvae xenograft model closer to the regulated preclinical drug discovery path.

## Figures and Tables

**Figure 1 pharmaceuticals-13-00001-f001:**
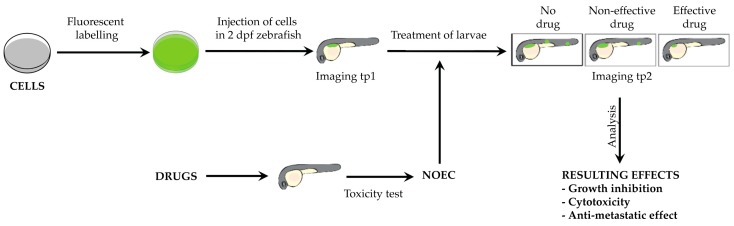
ZeOncoTest pipeline. Fluorescently labeled cells are injected in 48 hpf larvae, subsequently imaged at tp1. After the first imaging, xenotransplanted larvae are incubated at 35 °C with the NOEC of candidate drugs, previously calculated. Subsequent imaging is performed at tp2. The evaluation of drug impact on tumor growth and metastatic potential is given by the calculation and comparison of tumor mass and dispersion of cells at the 2 time points, in treated and untreated animals.

**Figure 2 pharmaceuticals-13-00001-f002:**
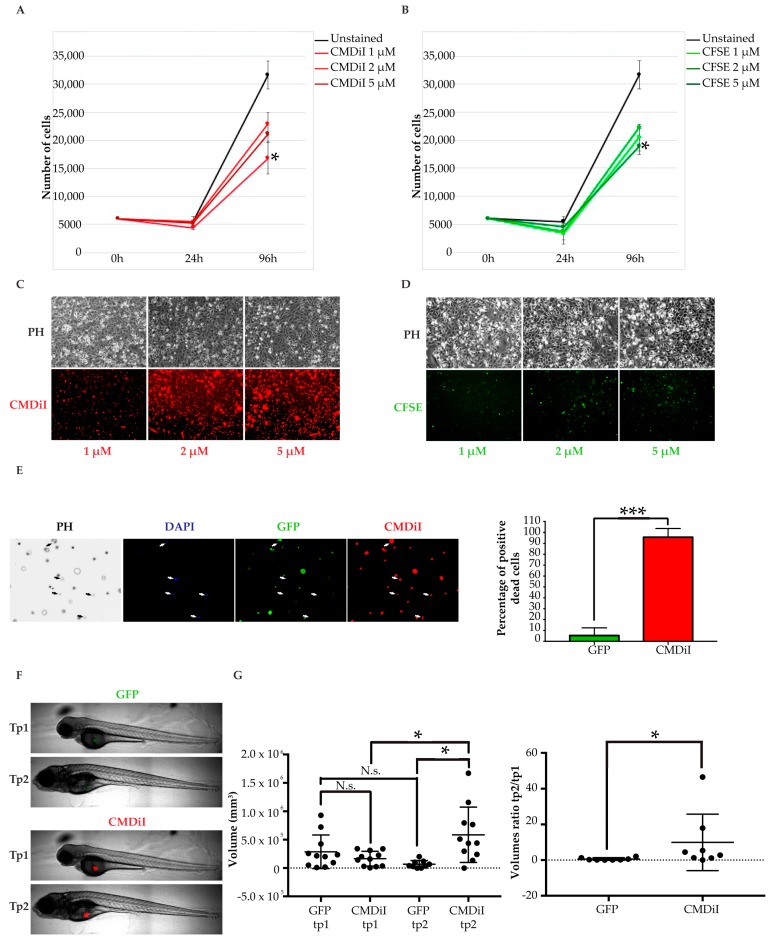
Fluorescent cell labelling methods coparison. (**A,B**) analysis of the toxicity of fluorescent dyes. CMDiI (**A**) and CFSE (**B**) dyes are toxic at different concentrations in MDA-MB-231 (growth curves). Mean and standard deviation of three counts are represented for every condition at each time point; (**C, D**) evaluation of cell staining homogeneity of fluorescent dyes. Representative images of cells stained with CMDiI (**C**) and CFSE (**D**) at day 4; (**E**) assessment of fluorescence retention in dead cells. MDA-MB-231 cells stably expressing GFP were stained with CMDiI and exposed to a high dose of DMSO. Dead cells were detected through DAPI staining. They are indicated by white arrows in the representative pictures on the left and quantified in the graph on the right. Six fields of view were analyzed for quantification. Results are represented as mean +/− standard deviation; (**F**) representative images of GFP+ and CMDiI+ tumors in the same injected fish at the two time points; (**G**) measurements of tumor volumes corresponding to the same GFP expressing cells and CMDiI labelled cells transplanted in each larva, at tp1 and tp2 (graph on the left). GFP+ and CMDiI+ tumor masses ratios between tp2 and tp1 (graph on the right). Each dot in the graphs represents the measurement of a fish. * indicates *p* < 0.05, ** means *p* < 0.01, 3 *** signify *p* < 0.001, and **** stays for *p* < 0.0001. No statistical significance is indicated by ns.

**Figure 3 pharmaceuticals-13-00001-f003:**
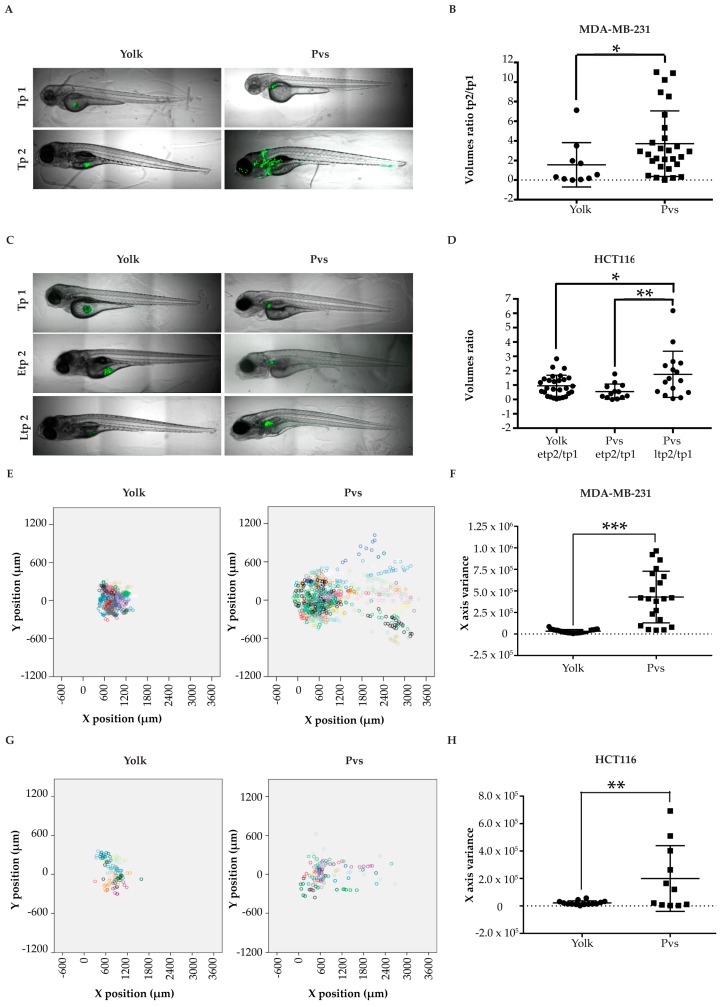
Injection sites comparison. (**A**) representative images of larvae injected with MDA-MB-231 cells in the yolk and in the pvs, at two different timepoints; (**B**) scatter dot plot of the tumor masses ratios between tp2 and tp1, for each larva. Tp1 corresponds to 2 hpi and tp2 to 96 hpi; (**C)** representative images of larvae injected with HCT116 cells in the yolk and in the pvs, at three different timepoints; (**D**) scatter dot plot of the volume ratios etp2/tp1 and ltp2/tp1 for each larva. Tp1 corresponds to 24 hpi, etp2 to 96 hpi and ltp2 to 144 hpi. (**E**,**G**) Dot plots showing MDA-MB-231 (**E**) and HCT116 cells (**G**) dissemination in injected larvae in the yolk and pvs at tp2. Each larva is depicted by a different color. Each dot corresponds to the location of a given segmented tumor mass related to the position of zebrafish eye. (**F**,**H**) scatter plots of tumor masses x variance at tp2 in larvae injected in the yolk and in the pvs with MDA-MB-231 (**F**) and HCT116 (**H**). Each dot or square in the graphs represents the measurement of a fish. * indicates *p* < 0.05, ** means *p* < 0.01, 3 *** signify *p* < 0.001, and **** stays for *p* < 0.0001. No statistical significance is indicated by ns.

**Figure 4 pharmaceuticals-13-00001-f004:**
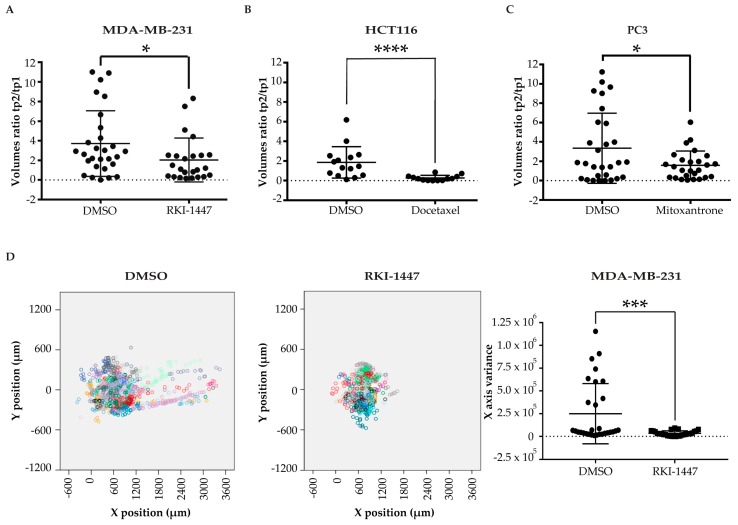
Pharmacological validation (**A–C**) scatter plots of the ratios between the tumor volumes at tp2 and tp1 of MDA-MB-231 (**A**), HCT116 (**B**), and PC3 (**C**) injected cells treated with RKI-1447, docetaxel, and mitoxantrone, respectively, vs. DMSO control. Tp1 refers to 2 hpi for MDA-MB-231 and 24 hpi for HCT116 and PC3. Tp2 corresponds to 96 hpi for MDA-MB-231 and PC3 and 144 hpi for HCT116; (**D**) combined scatter plots, on the left, and scatter dot plot, on the right, of the variance in the *x*-axis at tp2 of secondary tumor foci in MDA-MB-231 injected larvae, treated with control DMSO and RKI-1447. Each dot or square in the graphs represents the measurement of a fish. * indicates *p* < 0.05, ** means *p* < 0.01, 3 *** signify *p* < 0.001, and **** stays for *p* < 0.0001. No statistical significance is indicated by ns.

**Figure 5 pharmaceuticals-13-00001-f005:**
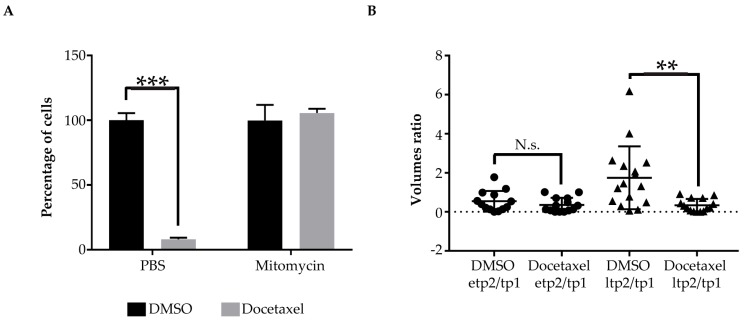
Addressing drugs MoA (**A**) bar graphs showing percentages of HCT116 cells surviving after incubation with mitomycin C or PBS control and subsequent treatment with docetaxel or DMSO control. Three cell counts were performed per condition. Results are represented as mean +/− standard deviation; (**B**) scatter dot plot of the tumor volume ratios between etp2 and tp1 and ltp2 and tp1 of HCT116 xenografts treated with Docetaxel or DMSO control. Tp1 corresponds to 24 hpi, etp2 to 96 hpi and ltp2 to 144 hpi. Each dot or triangle in the graphs represents the measurement of a fish. * indicates *p* < 0.05, ** means *p* < 0.01, 3 *** signify *p* < 0.001, and **** stays for *p* < 0.0001. No statistical significance is indicated by ns.

**Table 1 pharmaceuticals-13-00001-t001:** NOEC of selected drugs.

Drug	NOEC
Docetaxel	10 µM
Mitoxantrone	3 µM
RKI-1447	10 µM

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
