# Peer review of "ZeOncoTest: Refining and Automating the Zebrafish Xenograft Model for Drug Discovery in Cancer"

_pharmaceuticals, 2019, doi:10.3390/ph13010001_

Round 1
Reviewer 1 Report
This paper reports on the development of a new model for oncology testing using zebrafish xenograft larvae as animal model for cancer behavior and therapy guideline screening. This is an excellent model due to the lack of an efficient immune system which make injected cancer cells to not be rejected, allowing the formation of tumors. Therefore, this is an innovative approach presenting interesting findings by testing labelling methods, injection sites as well as pharmacologically validating the model by treating transplanted zebrafish larvae with tumoral cell lines. Furthermore, the authors addressed the mechanism of action of an anti-cancer drug.
The data presented show enough sensitivity to different tumor cells supporting the use of this animal model to as a promising in vivo screening platform for new antitumoral strategies.
Although this novel and original manuscript present interesting findings, some doubts arose while reading the manuscript and minor changes are suggested:
Abstract:
The abstract should be reviewed to include a further description of the methodology used as well as the results obtained.Results:
The results are too extensive. Some of the details described should be moved to the material and methods section in order to understand what was done. Also, references should not be included in this section which should focus only on the results obtained as described in the instructions for authors.Discussion:
Line 341 – references for “previous studies” are missing. Line 355 – this should be included in the materials section.Material and methods:
Please indicate the type of cell used, i.e MDA-MB-231 is breast cancer cell line. Describe all the abbreviations used. As the methods are too simple, the inclusion of referenced work is needed. Otherwise, complete the methods with more information. Line 462 – what is the developer solution? Line 463 – The calibration curve was based on what? Line466 – Taking in consideration the number of animals used, replicates and concentrations, did the authors equated the use of pronase for chemically dechorionated animals? Line 474 – It would be of interest to include the results for the calculation of the NOEC as supplementary material. Line 477 – the results from the teratogenic analysis are missing. Line 480 – what was the concentration of DMSO used in the study?References:
Review the references. Some present the journal pages other don’t.Author Response
Please see the attachment

Reviewer 2 Report
The manuscript titled ‘ZeOncoTest: refining and automating the zebrafish xenograft model for drug discovery in cancer’ reports the development and validation of use of zebrafish larvae Xenografts for drug screening in oncology.
The manuscript attempted to decipher an interesting concept but some improvements are needed to make it publishable:
As such the novelty of the manuscript is limited in the current form. It is important to highlight how this manuscript is significantly different than the previously developed and published zebrafish xenograft models.
Although the manuscript is well written, authors somewhat failed in establishing the feasibility of this model as a regular tool. Especially, authors have proposed this as an alternative to the animal studies. Additional discussion is needed in this direction.
Since zebrafish larvae is a much simpler organism than the humans or rodents, how the complex network of mammalian endocrine and paracrine factors on tumors will be taken account of? Authors need to justify this.
Reviewer 3 Report
In the article, need to explain the detail of choosing what dpf of larvae using in the experiment. In the section of testing the labeling system, the experiment duration is less than a week; however, fluorescence protein turnovers. Therefore, longer time frame for dye stability needs to be examined. Please discuss the phenomenon of different injection sites correspond to different cancer cell lines. Why these three types of cell line? Especially when fish has no two of the cells (breast, prostate). Any rejection or apoptosis responds after xanograft while there are no breast and prostate in fish? How long are author expecting to see the responds and how to validate? Please validate. As for the drug treatment, how did author decide the concentration? Please explain with the results. Please describe the details of drug treatment. As for the drugs are universal to whole cells, please demonstrate the fish phenotype after treatment.Author Response
Please see the attachment

Reviewer 4 Report
This study is a valuable contribution to reinforce the role of the zebrafish xenograft model to study human cancer progression and metastasis in the context of the organism and to test and select candidate drugs that have been pre-selected in vitro before using rodent models. This may lower significantly the costs of cancer drug discovery while contributing directly to the 3Rs strategy in the usage of experimental animals.
The authors have made a thorough bibliographic review of the technique and have optimized a pipeline for testing human cell line xenografts with a suitable imaging approach. However, I believe that this strategy is short of some important controls.
Authors have tested a dye against GFP expression in the human cells and have shown that GFP was more reliable to detect human cells. This is clearly shown in Fig. 3A with a highly metastatic line where also injection in pvs is shown to be better than in the yolk sac. However, all other Figures only show GFP+ tumors that could be debris after engulfment by macrophages. I strongly suggest that authors complement their results by showing human cell integrity (using anti-MHC class I or anti-human mitochodria).
Also, apoptosis should be demonstrated in the cases of tumor shrinkage after drug exposure by using caspase or tunnel assays.
Another important concern is the fact that from the images shown and comparing to other studies, the amount of cells injected seems to be quite low and far from to the 200-400 injected cells reported by authors. This is very important as cells do not have all the same engraftment rates and the injected inoculum has to be enough to guarantee a successful tumor implantation. Perhaps this is a wrong perception coming from the fact that authors have presented very few GFP images and so it would be important to shown also the GFP fluorescent images associated with the drug experiments in addition to the scatter plots.
Round 2
Reviewer 3 Report
Please explain the reason when calculate the tp1/tp2 volume? Any biological meaning? Also, please indicate the timepoints of tp1 and tp2. At the experiment that testing the labeling. The author claimed that the fluorescence is not stable after a period of time. Therefore, how author validate the decrease of cancer cells were killed by drugs or just lost of fluorescence signals? The author claimed that the pvs injection is better than yolk; however, the tumor mass ratios are diverse in pvs injection. Any discussion or any reason to make this conclusion?Author Response
Please see the attachment

Reviewer 4 Report
After the first review, authors have substantially improved the manuscript by adding references, rephrasing/clarifying some parts and checking spelling.
Addressing now the author's rebuttal for the points raised in the first review:
Point 1 - The authors now provide additional references and evidence supporting that GFP is only present in viable cells. Due to the nature of the model, it would still be preferable to have a confirmation that labelled cells are indeed live human cells, as references refer to either autologous transplants or mouse xenografts using immunodeficient mice which implies very different cell dynamics and immune interactions. Also, a simple immunostaining is a fast experiment that would not delay that much the publication. However, I believe both the fibroblast xenotransplant and the new TdTomato experiment are enough to sustain that it is very likely that the observed fluorescence comes from live tumor cells and I suggest that the texts is rephrased to indicate plausability rather than certainty.
Point 2 - The authors claim that the experiment required to demonstrate that growth inhibition is biologically supported by apoptosis is beyond the intent of this paper which aims at providing a fast assay for drug candidate selection. I fully agree that the assay should be designed to work in high throughput as simple and as fast as possible. However, in order to validate it and as a proof-of-principle that it is a reliable assay, authors should provide in this paper further evidence on the biological cellular mechanisms involved in such growth inhibition to show that the effect of the drugs on the cell lines in vivo is the same as reported in vitro.
Point 3 - The fluorescent images presented by authors to the several treatments clarified this question on the number of cells but it would be good to include these images as supplementary data
Round 3
Reviewer 4 Report
The authors have addressed appropriately Points 1 and 3 of my previous review by including proposed changes in the manuscript.
Regarding Point 2, since the authors refer that they already have an agreement with the editor not to postpone the publication, I rest my case although I consider that this stays as a major weakness on the scientific validation of this assay.